# Accelerated Weathering and Soil Burial Effect on Biodegradability, Colour and Textureof Coir/Pineapple Leaf Fibres/PLA Biocomposites

**DOI:** 10.3390/polym12020458

**Published:** 2020-02-16

**Authors:** Ramengmawii Siakeng, Mohammad Jawaid, Mohammad Asim, Suchart Siengchin

**Affiliations:** 1Department of Mechanical and Process Engineering, The Sirindhorn International Thai-German, Graduate School of Engineering (TGGS), King Mongkut’s University of Technology North Bangkok, Bangkok 10800, Thailand; ramengmawii8@gmail.com; 2Institute of Tropical Forestry and Forest Products (INTROP), Universiti Putra Malaysia, Seri Kembangan 43400, Malaysia; khanfatehvi@gmail.com

**Keywords:** biodegradability, accelerate weathering, soil burial, polylactic acid, coir fibres, pineapple leaf fibres

## Abstract

Accelerated weathering and soil burial tests on biocomposites of various ratios of coir (CF)/pineapple leaf fibres (PALF) with polylactic acid (PLA) were conducted to study the biodegradability, colour, and texture properties as compared with PLA.The biodegradability of a lignocellulosic composite largely depends on its polymer matrix, and the rate of biodegradation depends on many environmental factors such as moisture, light(radiation), temperature and microbes. Biodegradation was evaluated by soil burial and accelerated weathering tests. Changes in physical and morphological properties were observed in the biocomposites after weathering. These results allowed us to conclude that untreated CF/PALF/PLA biocomposites would be a more favourable choice owing to their better biodegradability and are suitable for the suggested biodegradable food packaging applications.

## 1. Introduction

Environmental concerns and awareness have paved the way to the development of biodegradable composites as a replacement for petroleum-derived or non degradable polymers. So, there is an increase in demand for natural fibre-based composites for commercial use in various industrial sectors [1]. A variety of biopolymers such as polylactic acid (PLA), polyhydroxyalkanoates (PHAs), and polybutylene succinate (PBS) are reported to be used as matrixes in composites. These biopolymers are naturally sourced and can potentially be combined with various natural fibres/lignocellulosic materials to produce biodegradable composites [2]. Natural fibres are sustainable materials in nature with advantages like lowcost, lightweight, renewability, and, most importantly, biodegradability [3,4].

The agricultural sector generates enormous amounts of agro-waste every year [5]. The lignocellulosic residues alone exceed 350 million tons per year and are poorly managed [6]. Recycling natural fibres by incorporating them into composites to manufacture renewable and biodegradable materials might help in waste reduction. This approach paves the way to the development of low-cost and biodegradable materials with promising characteristics [1]. This has also led to a considerable change in the research direction of fibre-reinforced polymer composites [7,8]. Natural-fibre-based biocomposites have been developed not only as a motivating factor for material scientists, but also as an opportunity to improve the lives of people globally by developing renewable and sustainable products. At present, plastic build up is a concerning matter due to its long-term environmental burden. A number of studies have shown that the seas and oceans are filled with plastic debris [9,10] that affects more than 250 aquatic species [11]. Plastic degradation can be achieved by multiple processes such as heat/thermic reaction, light/photo-oxidative reaction, ultraviolet (UV) degradation, etc. [12]. Polymeric substances usually degrade under UV radiation [13]. UV radiation induces degradation/modification of the surface chemistry in the composites, commonly known as photo-degradation or photo-catalysis [14]. The time required for their complete disintegration varies from one to another depending on the properties of the polymer, the type of light exposure, or the oxidation period. The degradation period could be a couple of months or in the range of hundreds to thousands of years [15,16,17].

PLA is one of the most studied biopolymers as it has many unique characteristics, including good transparency, glossy appearance, high rigidity, and good processability. PLA is frequently used for packaging materials. However, there are some limitations, notably, its inherent brittleness and poor toughness, slow degradation rate, and high cost, which hinder its extensive application [18,19]. Even though there are many limitations due to its material properties, a number of these challenges are expected to be overcome through blending PLA with other polymers, reinforcing it with fibres or fillers and additives, etc. [2]. A proper blend of coir fibres (CF)/pineapple leaf fibres (PALF) in a PLA matrix can possibly be used to develop a hybrid biocomposite which can match the thermo-mechanical properties of synthetic polymers and fibre-based composites which are being used in the manufacturing of food packaging materials [20]. With advancements in biocomposite technology, reinforcement of biopolymers/polymers with natural fibres (CF/PALF) is a method that can balance the harm caused by synthetic plastics and polymers. This will ensure environmentally friendly food packaging materials with high barrier properties and biodegradability [21]. It is intended that the use of CF/PALF/PLA hybrid composites will contribute to sustainability and environmental load reduction by the substitution of conventional food packaging materials.

Natural fibre/polymer composites are subjected to photo radiation when exposed to direct sunlight; this disrupts the chemical bonding of polymers, causing colour fading and disintegration with greater reductions in wetter conditions, especially in humid areas with high microbial activity [22,23,24]. After weathering/ageing periods, the strength of composites declines due to degradation of the fibres and polymers. Ageing in lignocellulosic fibres occurs due to the absorption of UV-rays by lignin present in the fibre, the formation of quinoid structures, Norrish reactions, and photo yellowing which occurs in lignin [25,26,27].

The biodegradability of composite materials can be analysed through different methods such as natural and accelerated weathering tests or soil burial tests [28] in normal garden soils or composts, hydrothermal degradation, chemical degradation and microbial attack, etc. Natural ageing or decaying processes are influenced by natural elements or environmental conditions. The long-term performance of composites in an exposed environment is evaluated by real-time observations for a period of a number of years [29]. A two-year ageing study on jute/phenolic biocomposites was conducted underexposure to natural weathering. Observed polymer cracking, black spots, bulging, fibrillation, and decline in tensile strength (over 50%) were reported [30]. Ochi [31] investigated the biodegradability of kenaf-reinforced PLA composites by composting for four weeks using a garbage-processing machine and reported a 38% decreased in the composite weights. Chee et al. [32] studied the effects of soil burial and accelerated weathering on the thermal and biodegradability properties of bamboo/kenaf-reinforced epoxy hybrid composites. They reported that soil burial showed more prominent degradation as compared to accelerated weathering.

Accelerated weathering tests are conducted in weathering chambers that mimic natural environmental conditions and the harsh effects of prolonged outdoor exposure. The method is carried out by exposing composite samples to UV radiation and controlled moisture/humidity and temperature. Accelerated weathering testing is a much faster and more convenient alternative method to natural weathering processes, and in addition, it is reproducible method. Accelerated weathering tests were performed on a kenaf/HDPE composite by Umar et al. [33] to test its durability. They observed micro-cracking of the surface and reduced tensile property and concluded that the biodegradability of composites was better enhanced by natural fibre reinforcements. Similar work was done with coir fibre [34], ramie, flax, cotton [35], hemp [14], kenaf, rice husk [23], and others, where the biodegradability was enhanced by increasing fibre loading in the composites. PLA is easy to process, biocompatible, and biodegradable in natural environments such as normal soil, compost, and aqueous medium [36,37,38]. The most important quality of PLA/NF composites is acceptable biocompatibility and biodegradability [39]. Yusoff et al. [40] studied three types of hybrid biocomposites based on PLA biopolymer, namely, kenaf/coir/PLA, bamboo/coir/PLA, and kenaf/bamboo/coir/PLA hybrid biocomposites. They concluded that these hybrid composites are suitable for indoor structural applications while maintaining their biodegradability at the time of disposal.

To date, limited research has been done on biodegradability and the impact of environmental conditions on hybrid biocomposites. In the present study, a novel hybrid biocomposite comprising CF and PALF with PLA biopolymer was developed and proposed for food packaging applications, particularly food trays. CF/PALF/PLA hybrid composites provide an option of achieving a blend of properties such as stiffness, degradability, processability, and strength, which are the core necessities of food packaging materials. The impacts of various environmental conditions on the degradation and physical properties of the hybrid composites were analysed through accelerated weathering and soil burial tests. The hybrid biocomposites were fabricated by a melt-blending and compression moulding method, with 30% fixed fibre loading by weight for different weight ratios of CF to PALF.

## 2. Materials and Methods

### 2.1. Materials

The PLA biopolymer used in this research was a general-purpose virgin pallet purchased from TT Biotechnologies Sdn. Bhd., Penang, Malaysia. The characteristics of PLA are shown in Table 1. The CF used in this research were supplied by Innovative Pultrusion Sdn Bhd.,Seremban, Malaysia, while PALF were procured from the southern part of India. The basic information regarding these fibres’ properties isdisplayed in Table 2. Sodium Hydroxide (NaOH) flakes used for fibre treatment were supplied by Evergreen Engineering & Resources, Selangor, Malaysia.

### 2.2. Fibre Treatment

CF and PALF were immersed in alkaline solution (6% NaOH) for 3 h. After immersion, these fibres were washed with tap water to attain neutral pH. Washed fibres were air dried under room temperature for 48 h. The methods relating to chemical selection, chemical concentration, and soaking time for this surface treatment were adopted from our previous published paper [41].

### 2.3. Composites Fabrication

CF and PALF were chopped into short fibres using a ring flaker machine; different fibre lengths were separated by mechanical sieves, and 1–2 mm length fibres were utilised for fabricating the composites. PLA pellets and fibres were dried at 80 °C for 24 h in a hot air oven.CF/PALF/PLA biocomposites were prepared by melt blending in an internal mixer at 180 °C temperature and 50 rpm speed for 10 min. The fibre loading was fixed at 30 wt% with different CF-to-PALF ratios (3:7, 1:1, and 7:3). The different weight percentages of CF and PALF in the PLA hybrid composites are shown in Table 3. Two different batches of each combination were prepared to verify reproducibility. The hybrid composites were compression moulded in a heated press at 180 °C for 10 min. After compression moulding, the samples were quenched using a water-cooled press and stored at room temperature before subsequent characterisation. Developed hybrid composites were cut into different shapes and sizes according to the respective characterisations and standards.

### 2.4. Accelerated Weathering Test

Accelerated weatheringwas carried out in an accelerated weathering chamber (Model QUV/Spray) in accordance with ASTM G 154-16 (2016). A UVA fluorescent bulb with 0.68 W/m^2^ irradiance at 340 nm wavelength was used with cycles of 1 h UV irradiation, followed by 1 min spray with de-ionised water and a subsequent 2 h condensation while maintaining a temperature/RH of 50 °C/55% and above. Samples of sizes 5″ × 3″ × 0.13″ were subjected to the weathering process for a duration of 250 h in the ageing chamber. The changes in the surface colour, surface texture, and weight of the tested samples were evaluated after 250 h of weathering and then compared with the unweathered samples.

#### 2.4.1. Weight Change

The weight change of the accelerated weathered samples was calculated and evaluated by the following equation:(1)WeightLoss(%)=[W0−W1/W0]x 100
where W0 and W1 are the weights of the sample before and after accelerated weathering, respectively.

#### 2.4.2. Colorimetric Colour Analysis

The surface colour of the biocomposites were measured using chroma-meter (Konica Minolta Colorimeter, Bangi, Malaysia) according to the CIE *L***a***b** colour system by Lab Colour Space. The lightness (*L**) and two chromaticity coordinates (*a** and *b**) were measured for independent specimens at four different positions on each.

#### 2.4.3. Surface Texture Measurement

Surface texture measurement was carried out in order to investigate the change in roughness of the samples’surfaces to evaluateof the degree of degradation by accelerated weathering. The surface texture (roughness) was measured usinga surface roughness tester (Surftest SV-3100 Bangi, Malaysia) in measuring conditions of measuredlength(X), 17.5000 mm; measured pitch, 0.0010 mm; Z1-axis range, 0.0800 mm; stylus radius compensation, 0.000000 mm; and Z gain adjusted ratio of 1.005017 by the X-axis pitch sampling method with nonpolar reversal and a measuring speed of 1.00 mm/s.

### 2.5. Soil Burial Test

Biodegradation tests on the CF/PALF/PLA biocomposites and neat PLA samples were carried outby a simple soil burial test, ASTM D570-98 (1998), to simulate the natural biodegradation of biocomposites. Dog-bone-shaped samples were prepared according to ASTM 638-14 (2014) and were buried in natural soil in a plastic bag without any enzyme activity or any composting materials. They were located outdoors with an average temperature and relative humidity of 30 °C and 80%, respectively. The buried samples were removed at regular intervals (30, 60, 90, 120, and 150 days) for different characterisations. After each interval of soil burial, the samples were washed thoroughly with water to remove the soil debris from the surface of the samples. Samples were airdried at room temperature until theyreached a constant weight. The biodegradability was evaluated by measuring and comparing the weight change (loss) before and after burial. The weight losses of buried samples were calculated using the following Equation (1). In addition, the surface appearances of the buried samples were evaluated by using Image Analyser for visual comparison.

## 3. Results

### 3.1. Accelerated Weathering: Weight Change

Figure 1 illustrates weight change against exposure time for the biocomposites and PLA samples. It is clear that all samples gained weight up to some extent which was then lost after drying due to the biodegradation of the biocomposites. The weight gain was due to the hydrophilic properties of CF and PALF [42]. The weight loss was less than 0.5% for neat PLA. The calculated weight loss was less than 2% for untreated C30 and less than 1% for untreated P30 biocomposite samples, while it was about 1.25% for all the untreated hybrid CF/PALF/PLA samples. Furthermore, the weight loss was less than 1% for all alkali-treated biocomposites and hybrid biocomposites. Thus, it was established that mercerisation of both CF and PALF makes them more stable in comparison with untreated CF- and PALF-based biocomposites. This may be due to enhanced interfacial adhesion between the fibres and matrix and/or improved interfacial bonds caused by improved hydrophobicity in biocomposites fabricated with alkali-treated fibres [43]. The calculated weight loss values were highest for the untreated C7P3 among the hybrid biocomposites, containing 70% untreated CF and 30% untreated PALF, owing to the high content of CF which is a lignin-rich natural fibre.

It can be seen from Figure 2 that the neat PLA sample remained nearly the same, without any visible discolouration or change in its original shape, whereas the biocomposites crumbled, forming fractures, cracks, and holes on the surface. The percentage weight loss in all the biocomposites were in a linear relationship with the progression of degradation. Similar observations were reported by Umar et al. [44] of the effect of accelerated weathering on kenaf/HDPE composites. According to Mehta et al. [42], these fractures, cracks, and holes were produced by the degradation of fibres. Nonetheless, differences in the degree of degradation were observed among the biocomposites with and without alkali-treated fibres. That is, the loss of shape or warping occurred slower in the biocomposites with alkali-treated fibres as compared to untreated CF/PALF/PLA biocomposites. We assume that this happened due to improved interfacial bonding between CF/PALF and PLA in the alkali-treated biocomposites, owing to the reduced hydrophilicity of alkali-treated fibres.

### 3.2. Accelerated Weathering: Surface Roughness

Surface texture measurements were carried out in order to investigate the change in roughness of the tested surfaces to evaluate the degree of degradation by accelerated weathering.

Figure 3 shows the change in surface roughness parameter Ra before and after accelerated weathering for untreated and alkali-treated CF/PALF/PLA biocomposites as well as neat PLA against exposure time, i.e., 250 h. Generally, the textures of the biocomposite samples became rougher with increasing weathering time [42]. Ra had the lowest values among all the surface roughness parameters, followed by Rz and Rmax with the highest values, as displayed in Table 4. The roughness parameters of the sample C30 A were the highest, followed by P30 A, followed by the C3P7A and C1P1A hybrid biocomposites. The untreated C7P3 hybrid biocomposite sample showed the lowest values of surface roughness parameters. In general, alkali-treated fibre biocomposites had higher surface roughness compared to untreated biocomposites. The C3P7A hybrid biocomposite sample showed the highest surface texture change among the hybrid biocomposites.

In the accelerated weathering chamber, the test samples were sprinkled with water to adjust the environment relative humidity, irradiated by UV-rays, and reacted with ambient oxygen [25,26]. These environmental conditions lead to breakage of the bonds between the fibres and the polymer matrix, which caused the biocomposite surfaces to crumble [42]. This spread through the thickness of the biocomposites over the exposure time via various reactions and processes such as oxidation, reduction, hydrolysis, dehydration, matrix crystallisation, swelling, shrinking, freezing, cracking, and interfacial degradation [43,45]. The major constituents responsible for the absorption of moisture in the fibres are (i) hemicellulose, (ii) accessible cellulose, (iii) non-crystalline cellulose, (iv) lignin, and (v) crystalline cellulose, in descending order. The major constituents responsible for UV degradation are (i) lignin, (ii) hemicellulose, (iii) accessible cellulose, (iv) non-crystalline cellulose, and (v) crystalline cellulose, whilst hemicellulose, cellulose, and lignin are the constituents responsible for thermal degradation [45]. The biological degradation of fibre is influenced greatly by hemicelluloses, followed by accessible cellulose and non-crystalline cellulose [42,46].

### 3.3. Accelerated Weathering: Colour Change

The Lab colour space is a colour-opponent space with dimension “*L*” for lightness and “*a*” and “*b*” for the colour opponent dimensions. Inthe Lab scale, “***L***” is light to dark, “***a***” is green to red, and “***b***” is blue to yellow [47].

Figure 4 illustrates the cumulative colour changes (dE) in the biocomposite samples with exposure time. According to the CIE*L***a*b** colour system, there are three parameters for colour: lightness (*L**) and two chromaticity coordinates (*a** and *b**). Here, +*a** to −*a** denotes red and green, +*b** to −*b** is for yellow and blue, and *L** varies from black (0) to white (100). The value changes in *L**, *a**, and *b** were used to determine the cumulative colour change of the samples, dE* [48]. On average, the d*L* values of all the untreated biocomposites increased from 25 at 0 h exposure to 70 for 250 h exposure. This was due to the change in biocomposite colour from medium and dark brown to greyish white after UV irradiation and exposure to moisture, condensation, and relative humidity. On the other hand, the neat PLA sample showed decreased d*L* values from 82.34 to 58.65 after the weathering test. This was because the sample changed from transparent to a milky white colour due to a reduction in the transparency of the polymer [49]. After alkali treatment of the fibres, the d*L* values of composites increased from 25 to 65 on average, except for the P30 A sample, which showed an increase in d*L* from 51 to 71 as the colour of the PALF was already very pale prior to alkali treatment. The multiple action of moisture, oxygen, and UV-ray exposure induced the colour change of these biocomposite samples [42]. The change in *L* was less distinct for alkali-treated hybrid biocomposites such as C3P7A, C1P1A, and C7P3A which have both the lightness from PALF and darkness from CF in proportion. This indicates that fibres with alkali treatment retained a higher amount of hydrophobicity due to surface protection against photo-oxidation [43].

Increases in the values of da from 0 for all weathered samples were observed, whereas reductions in the values of dB from 0 to 250 h were recorded in all untreated biocomposites, hybrids, and neat PLA samples, but increased values were observed in alkali-treated C30 and C3P7 hybrid biocomposites. As shown in Table 5, the overall colour change parameter (dE) increased from ~40 to ~80 for all biocomposites except for the alkali-treated P30 and neat PLA samples, both of which had lighter colours prior to accelerated weathering. The dE values of untreated hybrid biocomposites such as C3P7, C1P1, and C7P3 were higher than those of alkali-treated hybrid biocomposites. A slightly higher colour change occurred in untreated P30 biocomposites compared to untreated C30 biocomposites.

### 3.4. Soil Burial: Weight Loss

It has been confirmed that PLA degrades naturally in soil, though it is less prone to degradation compared to other aliphatic biodegradable polymers exposed in a natural environment [50]. PLA degradation in soil is reported to occur by two processes:hydrolysis and conversion of lactic acid into gas and water. In the early stage of degradation, molecular chains of PLA are hydrolysed from higher to lower molecular weight, which can be accelerated by acids or bases. Moisture and temperature also affect this hydrolysis [51]. In this step, some bacteria and fungi in the soil catalyse the degradation via hydrolytic scission of ester groups into acid and alcohol, ultimately converting the lactic acid molecules into CO_2_, water, and biomass [52]. It is possible that the degradation generally proceeds from the interior of the samples and that the diffusion rate of degradation products is rather slow [34]. In contrast, soil microbes can easily degrade CF and PALF. As a result, these reinforcement fibres in the biocomposites would hasten the degradation process. In addition, CF and PALF can be easily reduced into simple biomass, causing minimal harm to the environment [50].

The biodegradability level of biocomposites was assessed by evaluating the weight loss (%) of material samples after 30, 60, 90, 120, and 150 days of soil burial. Figure 5a,b illustrates the percentage weight loss for neat PLA and for untreated and alkali-treated CF/PALF/PLA biocomposite samples. Neat PLA shows almost no weight loss, whereas the biocomposites show weight loss and gradual degradation with burial time. The percentage weight loss in all the biocomposites was linearly related with the number of days of soil burial. An approximate weight loss of 1.8% for neat PLA and weight losses of 15.2% and 18.6% for the C30 and P30 biocomposites (Figure 5a) were observed. The weight losses for successive alkali-treated CF- and PALF-reinforced PLA biocomposites were 6.9% and 8.4% (Figure 5b). Among the hybrid biocomposites, the highest weight loss was 16.8% by C3P7, while C7P3 and C1P1 had the same values overall. The weight losses for successive alkali-treated C3P7, C1P1, and C7P3 were 7.3%, 6.8%, and 6.9%, respectively. The higher weight losses of the untreated CF- and PALF-reinforced PLA biocomposites as compared to the alkali-treated CF- and PALF-reinforced PLA biocomposites may be attributed to poor fibre matrix adhesion, leading to faster degradation [53]. The alkali treatment led to reduced hydrophilicity in fibres and, hence, less moisture absorption by the biocomposites from the soil, which ultimately led to slower degradation of biocomposites with surface-modified CF and PALF [54]. The highest levels of weight loss (18.6% and 16.8%, respectively) took place concurrently in untreated CF- and PALF-reinforced PLA biocomposites after 150 days of soil burial.

It can be seen from Figure 6 that the neat PLA sample sheet remained almost the same, without any visible discolouration or change in its original shape. On the other hand, all the biocomposites crumbled and surfaces became rougher, forming fractures, cracks, and holes on the surface as degradation progressed owing to the degradation of fibres [55]. Nonetheless, the untreated and alkali-treated biocomposites achieved different degrees of degradation. That is, the change of shape or warping occurred more slowly in the composites with alkali-treated fibres compared to the untreated CF/PALF/PLA biocomposites, which we assumed was due to improved interfacial bonding in alkali-treated biocomposites owing to the reduced hydrophilicity in alkali-treated fibres. The results showed that biodegradation of PLA and CF/PALF/PLA biocomposites in the soil ecosystem is a complex process following different patterns due to uncontrolled biotic and abiotic factors. A similar trend of results was reported by Rudnik et al. [56] in their study of the degradation behaviour of PLA films and fibres by conducting soil burial in Mediterranean field conditions and laboratory simulation testing for 11 months. They concluded that the thickness and form of materials played a crucial role in the biodegradation process under soil burial and composting conditions. In general, CF/PALF/PLA biocomposite samples buried in soils degrade faster than those aged inside an accelerated weathering chamber. Morphological indications are presented in Figure 2 and Figure 6.

## 4. Conclusions

The biodegradability of a lignocellulosic composite largely depends on its polymer matrix, and the rate of biodegradation depends on many environmental factors, such as moisture, light (radiation), temperature, and microbes. Biodegradation was evaluated by soil burial and accelerated weathering tests. Changes in physical and morphological properties were observed in the biocomposites after weathering. It was found that the degradation rate of PLA was lower than that of CF/PALF/PLA biocomposites after 250 h of accelerated weathering. The effect of weathering on the morphology of the biocomposites was characterised by the formation of cracks and fractures, increased surface roughness, and colour and weight change as a result of degradation. Biocomposites degrade after weathering through photo-radiation, thermal degradation, oxidation, and hydrolysis. Water enhances the rate of degradation through the swelling of fibre, which leads to further light dispersion. The soil burial tests imply good biodegradability of the CF/PALF/PLA biocomposites. All the biocomposites showed weight loss and gradual degradation with burial time. The percentage weight loss in all the biocomposites was linearly related with the number of days of soil burial. C3P7 showed the highest weight loss among the hybrid biocomposites, i.e., 16.8%, while C7P3 and C1P1 had the same values overall. The weight losses for successive alkali-treated C3P7, C1P1, and C7P3 were 7.3%, 6.8%, and 6.9%, respectively. Thisdifference can be attributed to poor fibre matrix adhesion, leading to faster degradation. The alkali treatment led to reduced hydrophilicity in fibres and, hence, less moisture absorption by the biocomposites from the soil, ultimately leading to slower degradation of the biocomposites with surface-modified CF and PALF. From this study, we can conclude that untreated CF/PALF/PLA biocomposites would be a more favourable choice owing to their better biodegradability and are suitable for the suggested biodegradable food packaging applications.

## Figures and Tables

**Figure 1 polymers-12-00458-f001:**
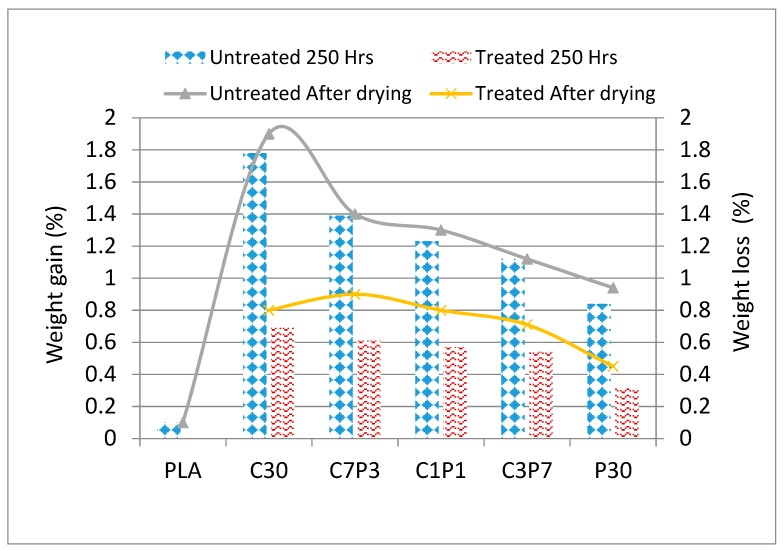
Change in weight over time for CF/PALF/PLA biocomposites, hybrid biocomposites, and PLA.

**Figure 2 polymers-12-00458-f002:**
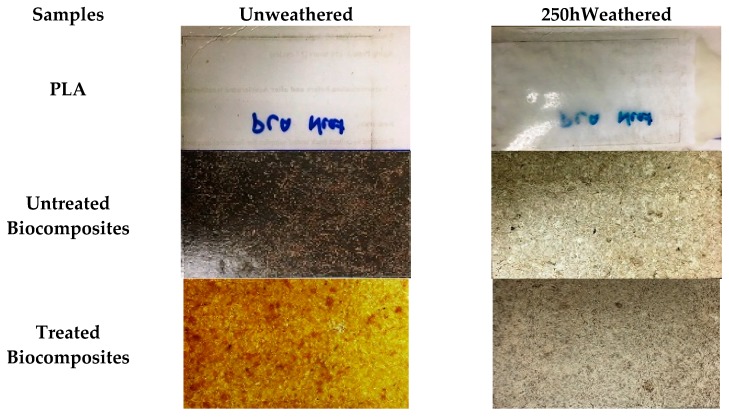
Images of unweathered and accelerated weathered untreated and treated CF/PALF/PLA biocomposites and PLA

**Figure 3 polymers-12-00458-f003:**
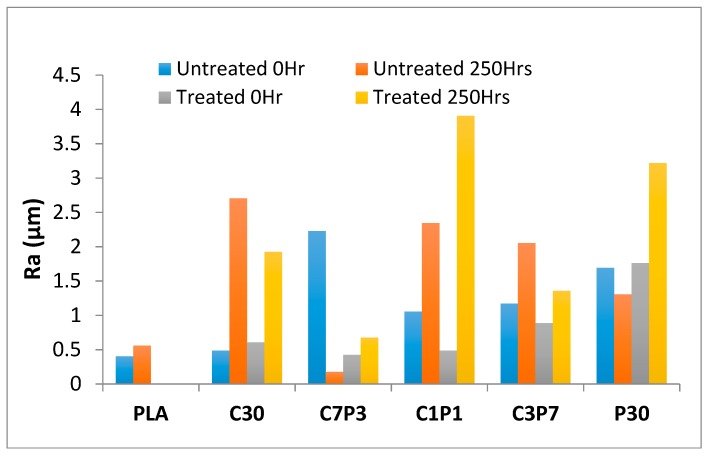
Change in surface roughness parameter Ra over time for PLA and untreated and alkali-treated CF/PALF/PLA biocomposites.

**Figure 4 polymers-12-00458-f004:**
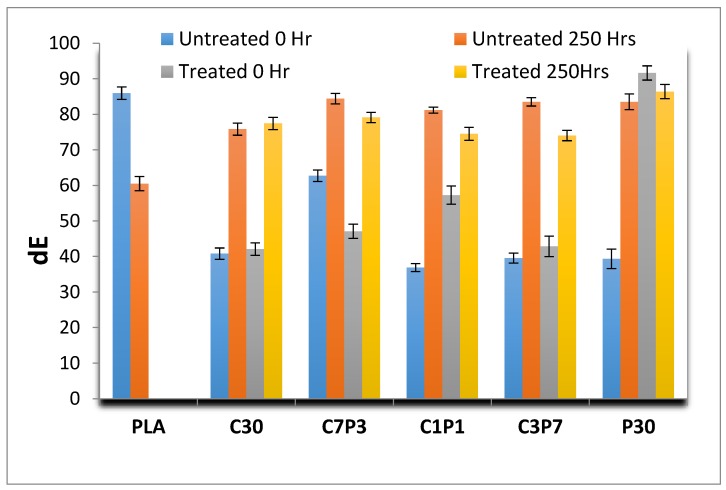
Cumulativecolour change (dE) over time for untreated and alkali-treated CF/PALF/PLA biocomposites.

**Figure 5 polymers-12-00458-f005:**
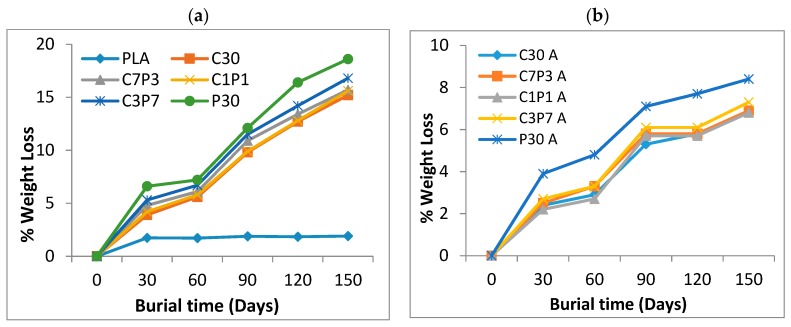
Weight loss (%) of (**a**) PLA and untreated CF/PALF/PLA biocomposites and of (**b**) alkali-treated CF/PALF/PLA biocomposites.

**Figure 6 polymers-12-00458-f006:**
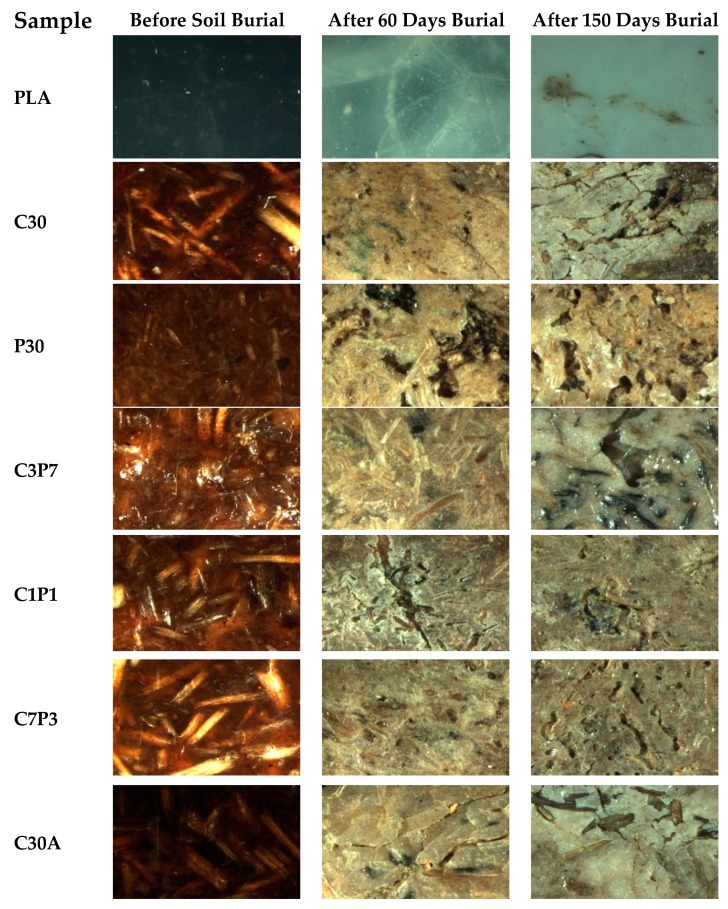
Image Analyser photographs of soil-degraded and unburied PLA sheets and CF/PALF/PLA biocomposites.

**Table 1 polymers-12-00458-t001:** Properties of polylactic acid (PLA) biopolymer [19].

Characteristics	Description	Characteristics	Description
Physical state	Pallets	Tensile strength	14.68 MPa
Colour	Translucent white	Young’s Modulus	3.98 GPa
Odour	No odour	Flexural strength	27.87 MPa
Density	1.25 g/mol	Flexural modulus	3.75 GPa
Melting point	150–170 °C	Impact strength	3.43 kJ/m^2^
Glass transition temperature	55–60 °C	Water absorption	2.58%
Molecular weight	74,000 g/mol	Thickness swelling	1.42%

**Table 2 polymers-12-00458-t002:** Properties of coir fibres (CF) and pineapple leaf fibres (PALF) [20].

Properties	CF	PALF
Density (g/m^3^)	1.20	1.07
Diameter (µm)	100–450	20–80
Cellulose (%)	42.14	81.27
Hemicelluloses (%)	15.17	12.31
Lignin (%)	35.25	3.46
Moisture absorption (%)	10.00	11.80
Microfibrillar angle (°)	30–45	8–15
Elongation at break (%)	17–47	1.6–4
Tensile Strength (MPa)	105–175	413–1627
Youngs Modulus (GPa)	4–6	34.5–82.5

**Table 3 polymers-12-00458-t003:** Formulation of CF/PALF/PLA biocomposites.

Designation	PLA (Weight %)	CF (Weight %)	PALF (Weight %)
Pure PLA	100	_	_
C-30	70	30	_
C7P3	70	21	9
C1P1	70	15	15
C3P7	70	9	21
P-30	70	_	30

**Table 4 polymers-12-00458-t004:** Surface texture measurements: roughness parameter summary sheet of untreated and alkali-treated CF/PALF/PLA biocomposites.

Sample	Before			After		
	Ra (µm)	Rmax (µm)	Rz (µm)	Ra (µm)	Rmax (µm)	Rz (µm)
PLA	0.4041	4.1468	2.9805	0.5611	5.8276	3.9426
C30	0.4878	18.0561	4.2805	2.7034	57.5719	23.4164
P30	1.6913	39.2247	19.9636	1.3076	20.4452	9.4624
C3P7	1.1726	30.6461	22.0001	2.0524	47.5884	18.2854
C1P1	1.0534	22.6111	14.3713	2.3452	34.8313	21.6659
C7P3	2.2285	23.7508	16.5913	0.1773	10.6666	3.0401
C30 A	0.6067	16.4739	6.0132	1.9237	66.6887	22.2321
P30 A	1.7618	13.5416	11.4903	3.2163	64.6825	35.9024
C3P7 A	0.8875	7.2143	3.3644	1.3577	62.3672	20.7893
C1P1 A	0.4864	16.4976	3.7916	3.9073	62.2191	36.5664
C7P3 A	0.4231	11.0194	4.7770	0.6759	28.9814	9.6606

**Table 5 polymers-12-00458-t005:** Colour change parameter summary sheet of PLA and untreated and alkali-treated CF/PALF/PLA composites using the CIELab system.

Sample	Before Weathering				After 250 h Weathering			
	*L**	*a**	*b**	dE	*L**	*a**	*b**	dE
PLA	82.34	0.52	3.07	85.93	58.65	1.29	0.55	60.49
C30	24.92	7.52	8.36	40.80	64.13	3.84	7.85	75.82
P30	23.27	7.75	8.33	39.35	75.48	1.27	6.76	83.51
C3P7	25.50	6.65	7.41	39.56	75.46	1.89	6.37	83.51
C1P1	23.56	6.44	6.89	36.89	73.64	2.03	5.51	81.18
C7P3	37.66	9.23	15.81	62.70	65.24	5.78	13.39	84.41
C30 A	25.02	7.65	9.40	42.07	63.23	4.40	9.79	77.42
P30 A	51.75	9.93	29.94	91.62	71.33	3.84	11.20	86.37
C3P7 A	28.29	6.51	8.06	42.86	60.15	3.69	10.18	74.02
C1P1 A	31.42	8.71	17.13	57.26	63.04	3.18	8.26	74.48
C7P3 A	28.78	7.54	10.76	47.08	66.54	3.58	8.98	79.10

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
