# Peer review of "Accelerated Weathering and Soil Burial Effect on Biodegradability, Colour and Textureof Coir/Pineapple Leaf Fibres/PLA Biocomposites"

_polymers, 2020, doi:10.3390/polym12020458_

Round 1

Reviewer 1 Report

Accelerated weathering and soil burial tests on various ratio of coir (CF)/pineapple leaf fibres (PALF) with polylactic acid (PLA) biocomposites were conducted to study biodegradability, colour and texture properties as compared with PLA. The biodegradability of a lignocellulosic composite largely depends on its polymer matrix and the rate of biodegradation depends on many environmental factors, such as moisture, light(radiation), temperature and microbes. Biodegradation was evaluated by soil burial and accelerated weathering tests. Changes in physical and morphological properties were observed in the biocomposites after weathering. These results conclude that untreated CF/PALF/PLA biocomposites would be more favourable choice owing to their better biodegradability and are suitable for the suggested biodegradable food packaging applications.

In Abstract and Conclusions sections, the content should be simplified and focused. Figure 1 is not very clear. Please improve resolution of the Figure. Please keep the decimal point consistent as the scientific reports. For example, “10 and 11.8 in Table 2”, “22, 14.3713, and 4.777 in Table 4”. In the paper, the unit writing mentioned must be uniform. For example, “3 hrs (L144)”, “24 h (L151)”. Figures 4, 6, 7, 8, 9, 10 should reflect statistical analysis data. “crystalline cellulose whilst hemicellulose, cellulose, and lignin are the responsible constituents for thermal degradation”. Some references are needed. There was an error in Figure 3. The expression PLA for P-30 should be 30, not 70. Please merge all the pictures in Figure 3 and Figure 10. At last, the level of English throughout this manuscript should be improved. There are a number of grammatical errors and instances of badly worded/constructed sentences. Please check the manuscript and refine the language carefully.

Author Response

Editor

Polymers

Submission: Accelerated Weathering and Soil Burial Effect on Biodegradability, Colour and Texture of Coir/Pineapple Leaf Fibres/ PLA Biocomposites

Dear Editor,

Thanks for referees’ valuable comments and suggestions for our manuscript. We have revised the manuscript accordingly, we have highlighted changes through track change for easy access. All the additions/ Modifications/ corrections are updated in the main manuscript. Whole manuscript is rechecked again to correct the grammatical errors and modified accordingly. A detailed corrections point by point are listed below Referees' Comments to Author:

1st Reviewer

Open Review

Comments and Suggestions for Authors

Thank you very much for giving your valuable comments and suggestions. We have corrected all the point, have been raised.

Accelerated weathering and soil burial tests on various ratio of coir (CF)/pineapple leaf fibres (PALF) with polylactic acid (PLA) biocomposites were conducted to study biodegradability, colour and texture properties as compared with PLA. The biodegradability of a lignocellulosic composite largely depends on its polymer matrix and the rate of biodegradation depends on many environmental factors, such as moisture, light(radiation), temperature and microbes. Biodegradation was evaluated by soil burial and accelerated weathering tests. Changes in physical and morphological properties were observed in the biocomposites after weathering. These results conclude that untreated CF/PALF/PLA biocomposites would be more favourable choice owing to their better biodegradability and are suitable for the suggested biodegradable food packaging applications.

Reply: we are agreed to revise the abstract and amendment is made in abstract accordingly

In Abstract and Conclusions sections, the content should be simplified and focused.

Reply: the abstract and conclusion section is rechecked and simplified accordingly.

Figure 1 is not very clear. Please improve resolution of the Figure.

Reply: the resolution of figure was not good, so we prefer to delete it.

Please keep the decimal point consistent as the scientific reports. For example, “10 and 11.8 in Table 2”, “22, 14.3713, and 4.777 in Table 4”.

Reply: all the decimal points have been rechecked and corrected.

 In the paper, the unit writing mentioned must be uniform. For example, “3 hrs (L144)”, “24 h (L151)”.

Reply: the unit has been checked and corrected accordingly, the new line no is L160

Figures 4, 6, 7, 8, 9, 10 should reflect statistical analysis data.

Reply: In manuscript, figures and table both were included, so we deleted the figures. All the figures are summerised in Table 5

 “crystalline cellulose whilst hemicellulose, cellulose, and lignin are the responsible constituents for thermal degradation”. Some references are needed.

Reply: Reference cited,  reference no (45)

 There was an error in Figure 3. The expression PLA for P-30 should be 30, not 70.

Reply: There was no mention of P-70 in Figue 3, may be reviewer mixed with some other figure

Please merge all the pictures in Figure 3 and Figure 10.

Reply: We reduced the number of figures, so we think, there is no need of merger.

At last, the level of English throughout this manuscript should be improved. There are a number of grammatical errors and instances of badly worded/constructed sentences. Please check the manuscript and refine the language carefully. 

Reply: Carefully checked and refine up to our best.

Reviewer 2 Report

General comments

In my opinion, the study is well designed and apparently it has been conducted in an appropriate manner. Nonetheless, its originality is medium-low (see comments below). The results are somewhat limited (measurements of tensile strenght and Young modulus are missing) and its interest is medium. It lacks conciseness, especially in the introduction (very lenghty), and there are several figures that are not necessary. A proper discussion, with comparisons with results from previous works, is missing.

Specific comments

Abstract

It is necessary to summarize it. There are repeated phrases (L19-21, L23-24, L26, L28-29)

L17. Colorimetric

Introduction

Too wordy. The 130 lines can be reduced to 90-100 without any problem.

Originality is low (L70-74)

L83-84 and Figure 1 can be deleted

L109-112 can be deleted.

Again, originallity is low (L117-119)

Results

L207. The sentence is truncated. Something is missing.

L232, Figure 3. Reduce size. It is difficult to distinguish the appearance of the untreated composite from the treated one after 250 hours. Same applies to figure 11 (differences after burial, between 60 days and 150 days).

L243. Fix cross-reference.

L250. Reduce size.

L273. Fix cross-reference.

L275. Figure 5 can be deleted.

L304. Figures 6, 7 and 8 can be deleted. The information is redundant with Table 5.

L324. Space missing.

L368. Reduce size and place next to each other, in parallel.

L385. A proper discussion is missing (see general comments above).

Conclusions

L395. Delete "These processes alter the chemical and physical properties of biocomposites."

L402. Replace "The higher weight loss of untreated CF and PALF reinforced PLA biocomposites as compared to alkali treated CF and PALF reinforced PLA biocomposites is attributed" with "The difference can be attributed to..."

Author Response

Editor

Polymers

Submission: Accelerated Weathering and Soil Burial Effect on Biodegradability, Colour and Texture of Coir/Pineapple Leaf Fibres/ PLA Biocomposites

Dear Editor,

Thanks for referees’ valuable comments and suggestions for our manuscript. We have revised the manuscript accordingly, we have highlighted changes through track change for easy access. All the additions/ Modifications/ corrections are updated in the main manuscript. Whole manuscript is rechecked again to correct the grammatical errors and modified accordingly. A detailed corrections point by point are listed below Referees' Comments to Author

2nd Reviewer

Comments and Suggestions for Authors

Thank you very much for accepting our manuscript to review and giving your valuable comments and suggestions. We have corrected all the point, have been raised.

General comments

In my opinion, the study is well designed and apparently it has been conducted in an appropriate manner. Nonetheless, its originality is medium-low (see comments below). The results are somewhat limited (measurements of tensile strenght and Young modulus are missing) and its interest is medium. It lacks conciseness, especially in the introduction (very lenghty), and there are several figures that are not necessary. A proper discussion, with comparisons with results from previous works, is missing.

Reply: Tensile strength and Young modulus was not measured (not possible due to the deterioration of samples). Introduction is shortened as per suggestion. Unnecessary Figures are eliminated to improve conciseness.

Specific comments

Abstract

It is necessary to summarize it. There are repeated phrases (L19-21, L23-24, L26, L28-29)

Reply: I agree with reviewer’s comments. We have Simplified and summarized for more concise content

L17. Colorimetric

Reply: it has been Corrected

Introduction

Too wordy. The 130 lines can be reduced to 90-100 without any problem.

Reply: we do agree with reviewer, so the sentences have been Shortened and reduced

Originality is low (L70-74)

Reply: sentence is improved

L83-84 and Figure 1 can be deleted

Reply: L83-84 and Figure 1 are deleted as per suggestion.

L109-112 can be deleted.

Reply: L109-113 are deleted and amended in L113

Again, originallity is low (L117-119)

Reply: sentence is improved

Results

L207. The sentence is truncated. Something is missing.

Reply: sentence is improved

L232, Figure 3. Reduce size. It is difficult to distinguish the appearance of the untreated composite from the treated one after 250 hours. Same applies to figure 11 (differences after burial, between 60 days and 150 days).

Reply: Size reduced for these images

L243. Fix cross-reference.

Reply: All the cross references are rechecked and edited

L250. Reduce size.

Reply: size is reduced as per requirement

L273. Fix cross-reference.

Reply: All the cross references are rechecked and edited

L275. Figure 5 can be deleted.

Reply: Figure 5 are deleted as per suggestion and the related texts are amended

L304. Figures 6, 7 and 8 can be deleted. The information is redundant with Table 5.

Reply: Figure 6,7 and 8 are deleted as per suggestion and the related texts are amended

L324. Space missing.

Reply: it is corrected as required

L368. Reduce size and place next to each other, in parallel.

Reply: the corrections have been made accordingly.

L385. A proper discussion is missing (see general comments above).

Reply: we have modified the discussion to make more clear to the reader with justifications.

Conclusions

L395. Delete "These processes alter the chemical and physical properties of biocomposites."

Reply: the sentence is deleted as per suggestion

L402. Replace "The higher weight loss of untreated CF and PALF reinforced PLA biocomposites as compared to alkali treated CF and PALF reinforced PLA biocomposites is attributed" with "The difference can be attributed to..."

Reply: the sentence is replaced as per suggestion

Round 2

Reviewer 1 Report

Based on the author's revision, this article could be published.

Author Response

thanks you very much for the acceptance 

Reviewer 2 Report

Thank you for addressing most of the issues raised.

Author Response

thank you for acceptance of all answers